# UDC: Unified DNAS for Compressible TinyML Models for Neural Processing Units

**Igor Fedorov**[*]
Meta AI
Menlo Park, CA 94025
`fedorov.uofi@gmail.com`

**Ramon Matas**[*]
NVIDIA
Santa Clara, CA 95050
`ramonm@nvidia.com`

**Hokchhay Tann**[*]
Tenstorrent
Boston, MA 01721
`ctann@tenstorrent.com`

**Chuteng Zhou**
Arm Inc.
Boston, MA 01721
`chu.zhou@arm.com`

**Matthew Mattina**[*]
Tenstorrent
Boston, MA 01721
`mmattina@tenstorrent.com`

**Paul N. Whatmough**[*]
Qualcomm AI Research
Cambridge, MA 02140
`pwhatmou@qti.qualcomm.com`

## Abstract

Deploying TinyML models on low-cost IoT hardware is very challenging, due to limited device memory capacity. Neural processing unit (NPU) hardware address the memory challenge by using model compression to exploit weight quantization and sparsity to fit more parameters in the same footprint. However, designing compressible neural networks (NNs) is challenging, as it expands the design space across which we must make balanced trade-offs. This paper demonstrates Unified DNAS for Compressible (`UDC`) NNs, which explores a large search space to generate state-of-the-art *compressible* NNs for NPU. ImageNet results show `UDC` networks are up to $3.35\times$ smaller (iso-accuracy) or 6.25% more accurate (iso-model size) than previous work.

## 1 Introduction

IoT applications demand *TinyML* models that fit on highly-constrained hardware (HW), with limited memory and compute power [14, 25, 30, 42]. Canonical TinyML tasks include visual wakewords, audio keywords, anomaly detection, speech enhancement, and image classification [14, 13, 26, 9, 48]. Traditional microcontroller units (MCUs) are not well suited to meet the memory and compute challenges of TinyML, so HW vendors offer specialized processors for neural network (NN) inference, called neural processing units (NPUs) [2, 1].

MCU inference runtimes [7, 8] do not implement model compression, which is slow in software. Hence MCUs do not benefit from sub 8-bit quantization or unstructured pruning [31]. In contrast, NPUs with HW model compression [2, 1] do benefit from both optimizations [3]. For example, the Arm Ethos-U55 NPU compiler (Vela [3]) encodes weight tensors using two components: 1) a binary mask marking non-zero elements (run-length and Golomb-Rice compressed), and 2) the non-zero values (Golomb-Rice compressed). A model aggressively pruned to have many zeros and quantized to low bitwidths can be deployed in very little device memory. At inference time, weights are then decompressed by the NPU on demand.

NPU compression is an enormous advantage for TinyML, where meager Flash memory limits model size and therefore accuracy [25, 48, 30, 42]. A smaller memory footprint also reduces memory access energy, critical for battery powered IoT devices [45, 11, 31, 5]. Fig. 1 shows

---

[*]Work done while at Arm Inc., Boston, MA

36th Conference on Neural Information Processing Systems (NeurIPS 2022).

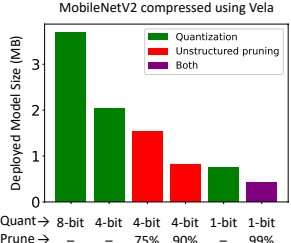 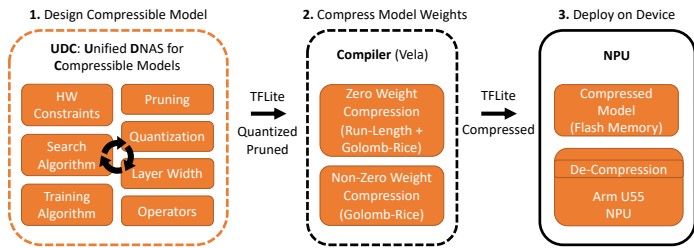

Figure 1: **Left:** NPU weight compression allows significant deployed model size reduction for TinyML. Compressed MBNetV2 size scales with increasingly aggressive quantization and pruning choices. **Right:** UDC designs compressible models, tailored to NPU deployment.

how model size scales with quantization, pruning, and both, when compressing a MBNetV2 [35] for deployment using Vela, down to a 9.3× reduction in model size for 1-bit quantization and 1% non-zero weights in the extreme.

To fully exploit NPU model compression, we define our problem statement as follows:

> **Automatically design compressible NNs with the smallest model size.** (P1)

Typically, the model architecture itself is developed first, either through manual trial-and-error or by neural architecture search (NAS) [47, 21, 68, 33, 54, 67, 26, 62, 65]. Then, model conditioning techniques, e.g., quantization [61, 69, 64, 66, 63] and unstructured pruning [18, 43, 32, 31, 50], are applied before deploying to the target HW. Furthermore, the generated model must also be small enough to deploy on the NPU device; merely regularizing size does not guarantee that the models found will fit into device Flash memory. It is essential that:

> **Generated NNs do not exceed a hard constraint on (compressed) model size.** (P2)

This paper describes UDC (Fig. 1), which merges the model design and conditioning steps by conducting a joint search over NN architecture, weight bitwidths, and sparsity rates. UDC builds upon differentiable NAS (DNAS), which exploits efficient weight sharing to solve (P1), while addressing key challenges like how to explore the design space while still addressing (P2) [51, 56]. The search space is HW-compression-aware, including only model conditioning techniques supported by the real-world Vela NPU compiler. We explicitly exclude low-rank matrix factorization [38] and non-uniform floating-point quantization [66, 63], which are not supported. The contributions of this paper are further summarized below.

**Joint network architecture and conditioning search** We extend the DNAS formalism to learn layer-wise weight sparsity levels. We present a method for searching for sparsity levels in conjunction with layer bitwidths. We show how to maximize weight sharing while jointly searching over sparsity, bitwidth, and layer width, as well as provide a differentiable and easily computable measure of compressibility for the DNAS algorithm to optimize.

**Novel search algorithm** Previous work (e.g. [22, 21]) fails to effectively trade-off accuracy with model size (P2) in our search space (Table 3, first row). UDC addresses this, with the following improvements: 1) guarantees that the search yields a model which satisfies specified HW constraints (Sec. 4), 2) provides control over exploration-exploitation (Sec. 4.1), and 3) avoids over-regularization from biased Gumbel-softmax approximation (Sec. 4, 4.2).

**Novel sparse, low-bitwidth representation and training algorithm** UDC yields compressible models that are difficult to train. We identify the root cause of the problem and propose a solution using a novel weight representation (Sec. 5).

**SOTA NPU-deployable TinyML models** Using UDC, we demonstrate compressible NNs Pareto dominant over prior work on CIFAR100, ImageNet, and super resolution (SR) tasks (Fig. 4) [41, 57, 10, 37]. Compression gains are validated by deploying to Ethos-U55 NPU.

## 2 Related work

Table 1 contrasts UDC with relevant previous work. MCUnet [48] is the closest to UDC, as it targets the TinyML HW form-factor with severely limited memory and includes results on large scale datasets like ImageNet. However, UDC is fundamentally different to MCUNet: 1) unlike MCUnet, UDC also performs per-layer bitwidths and sparsity rate search to exploit HW compression, 2) UDC is a DNAS algorithm whereas MCUnet uses evolutionary search of a pre-trained once-for-all (OFA) supernet [17]. In par-

Table 1: UDC features vs. related work.

| | MCUnet [48] | APQ [64] | HAQ [63] | Yang et al. [66] | Gong et al. [27] | Choi et al. [18] | Uhlich et al. [61] | FBNetV2 [62] | UDC (Ours) |
|---|---|---|---|---|---|---|---|---|---|
| Width/Operator/Depth | ✓ | ✓ | | | | | | ✓ | ✓ |
| Bitwidth/Sparsity | | | | | | | | | ✓ |
| HW Constraint Guarantee | ✓ | ✓ | | ✓ | ✓ | | ✓ | | ✓ |
| Deployable w/Integer Math | ✓ | ✓ | | | ✓ | ✓ | ✓ | ✓ | ✓ |
| TinyML Size ($< 1.25$MB) | ✓ | | ✓ | ✓ | ✓ | ✓ | ✓ | | ✓ |

ticular, the UDC search space targets compressible models and is much more diverse than that of MCUNet; training a single OFA supernet for our search space would be infeasible (Sec. 6.2).

Yang et al. [66] also demonstrate learning of per-layer bitwidth and sparsity rate, but otherwise differ significantly in that they: 1) do not include storage of the (required) pruning mask in the reported model size, which can even dominate at high sparsity rates [16]; 2) only consider model conditioning, without accompanying NN width, depth, or operator search; 3) employ floating-point non-uniform quantization, not deployable on TinyML MCU and NPU platforms which only support integer operations.

APQ proposes a multi-stage algorithm to search over layer width and bitwidth [64]. Compared to APQ, UDC: 1) optimizes over sparsity rates whereas APQ does not, and 2) produces models $3$–$9\times$ smaller than APQ. Other relevant works include Gong et al. [27], which searches over width and bitwidth but not sparsity, Choi et al. [18], which searches over bitwidth and sparsity but not layer widths and without addressing (P2), and Uhlich et al. [61] which searches over only bitwidths.

## 3 Modelling design decisions

We use bold-face to denote vectors/tensors, and $z[k]$ to mean the $k$'th element of vector $\mathbf{z}$. For a layer with input $\boldsymbol{x}$ and parameters $\boldsymbol{\theta}$, we denote its output by $f(\boldsymbol{x}, \boldsymbol{\theta})$. We say $\boldsymbol{z} \in \mathbb{R}^K$ follows a categorical distribution parameterized by $\boldsymbol{\pi}$, i.e. $\boldsymbol{z} \sim \mathrm{Cat}(\boldsymbol{\pi})$, if $p(z[k] = 1) = \pi[k]$.

**Width selection** Layer widths are modelled as $f(\boldsymbol{x}, \boldsymbol{\theta}) \odot \boldsymbol{w}, \|\boldsymbol{w}\|_0 / |\boldsymbol{w}| = \rho$, where $\boldsymbol{w}$ is a binary mask to toggle any given channel, $\rho$ is the fraction of non-zero channels, $\odot$ is element-wise multiplication, $\|\boldsymbol{w}\|_0$ is the $\ell_0$ pseudo-norm which counts the number of non-zeros in $\boldsymbol{w}$, and $|\boldsymbol{w}|$ is the number of elements in $\boldsymbol{w}$. We adopt the convention of setting the first $\rho$ fraction of the channels of $\boldsymbol{w}$ to 1 [62]. Different choices $\{\rho_1, \cdots, \rho_{K_w}\}$ of layer width can be modelled by the random variable (RV)

$$\boldsymbol{w}(\boldsymbol{\pi_w}) = \sum_{k=1}^{K_w} z_w[k]\boldsymbol{w_k}, \boldsymbol{z_w} \sim \mathrm{Cat}(\boldsymbol{\pi_w}), \|\boldsymbol{w_k}\|_0 / |\boldsymbol{w_k}| = \rho_k. \quad (1)$$

We refer to $\boldsymbol{z_w}$ as a decision variable and to $\rho_k$ as an option.

**Sparsity** Sparse weight tensors are expressed as $\boldsymbol{\theta} \odot \boldsymbol{m}, \|\boldsymbol{m}\|_0 / |\boldsymbol{m}| = s$ where $\boldsymbol{m}$ is a binary mask and $s$ is the fraction of non-zeros. We set the non-zeros of $\boldsymbol{m}$ to correspond to the largest magnitude elements of $\boldsymbol{\theta}$ [50, 31, 28]. We model different choices of $s$ amongst $\{s_1, \cdots, s_{K_s}\}$ with the RV

$$\boldsymbol{m}(\boldsymbol{\pi_s}) = \sum_{k=1}^{K_s} z_s[k]\boldsymbol{m_k}, \boldsymbol{z_s} \sim \mathrm{Cat}(\boldsymbol{\pi_s}), \|\boldsymbol{m_k}\|_0 / |\boldsymbol{m_k}| = s_k. \quad (2)$$

**Quantization** Uniformly quantized tensors are given by

$$Q\left(\boldsymbol{\theta}, b, r\right) = \begin{cases} d \times \text{round}\left(\frac{\text{clip}(\boldsymbol{\theta}, -r, r)}{d}\right), & r > 0, \ \ d = r/(2^{b-1} - 1) & b > 1 \\ \text{sign}\left(\boldsymbol{\theta}\right) & b = 1 \end{cases} \tag{3}$$

where $r$ is the quantization range and $b$ the bitwidth. We model different choices of $b \in \left\{b_1, \cdots, b_{K_q}\right\}$ as the RV $q\left(\boldsymbol{\pi_q}\right) = \sum_{k=1}^{K_q} z_q[k] Q\left(\boldsymbol{\theta}, b_k, r_k\right), \boldsymbol{z_q} \sim \text{Cat}\left(\boldsymbol{\pi_q}\right)$, where each bitwidth $b_k$ is parameterized by its own range $r_k$.

**Sparsity and quantization** Sparse, quantized tensors are modeled as $Q(\boldsymbol{\theta}, b, r) \odot \boldsymbol{m}$. We find setting the non-zero values of $\boldsymbol{m}$ based on $\boldsymbol{\theta}$ superior to using $Q(\boldsymbol{\theta}, b, r)$, which discards information about the relative magnitude of weights during the quantization process. To model tensors whose sparsity level and bitwidth must be chosen, we define the RV $q(\boldsymbol{\pi_q}) \odot \boldsymbol{m}(\boldsymbol{\pi_s})$.

**Operator selection** We frame the choice over different layer operators as the RV $\sum_{k=1}^{K_f} z_f[k] f_k(\boldsymbol{x}, \boldsymbol{\theta}_k), \boldsymbol{z_f} \sim \text{Cat}\left(\boldsymbol{\pi_f}\right)$, where each operator has a (possibly) different functional form $f_k(\cdot)$ parameterized by its own weights $\boldsymbol{\theta}_k$. By including identity as a candidate operation, we can also model varying NN depth.

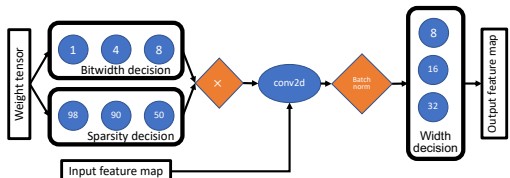

Figure 2: UDC search space overview.

Fig. 2 shows the layer-level search space, excluding operator selection for brevity. Weight sharing is maximized by using the same $\boldsymbol{\theta}$ in the sparsity, bitwidth, and width decisions. The output of the layer in Fig. 2 is modelled as the RV $f\left(\boldsymbol{x}, q(\boldsymbol{\pi_q}) \odot \boldsymbol{m}(\boldsymbol{\pi_s})\right) \odot \boldsymbol{w}(\boldsymbol{\pi_w})$.

## 3.1 Computing layer storage size

Satisfying (P1)-(P2) necessitates a quantitative measure of storage size, accounting for data compression. For a given layer, the storage size achievable by prefix-free compression is lower-bounded by the weight (empirical) entropy, $H(\cdot)$, times the number of weight elements [19]. The entropy bound can, in turn, be bounded by (Appendix C):

$$H\left(Q(\boldsymbol{\theta}, b, r) \odot \boldsymbol{m}\right) \times \|\boldsymbol{w}\|_0 \le \left(b + \left(-s \log_2 s - (1-s) \log_2 (1-s)\right)\right) \times \|\boldsymbol{w}\|_0. \tag{4}$$

The advantage of the right hand side (RHS) of (4) over the left hand side (LHS) is that it can be computed cheaply without processing $\boldsymbol{\theta}$. Using the LHS requires computing the empirical entropy, which is expensive for large $\boldsymbol{\theta}$. Moreover, combining the LHS with a gradient-based optimizer requires gradient approximation, since the empirical entropy is not differentiable [60]. We refer to the RHS of (4) as the compressed tensor size and use it as our measure of layer size. Sec. 6.2 confirms that it is achievable with both an arithmetic encoder and Vela. Let $\epsilon(s, b, \rho)$ be a given layer's storage size (the RHS of (4)) as a function of sparsity $s$, bitwidth $b$, and non-zero channel fraction $\rho$. When $(s, b, \rho)$ must be chosen, let $\epsilon\left(\sum_k^{K_s} z_s[k] s_k, \sum_k^{K_q} z_q[k] b_k, \sum_k^{K_w} z_w[k] \rho_k\right)$ be the storage size. For the entire NN, we sum the storage size of all layers and denote the result $\mathcal{E}\left(\{\boldsymbol{z}\}\right)$, where $\{\boldsymbol{z}\}$ is short-hand for the set of all decision variables.

## 4 Proposed DNAS algorithm

Our optimization objective is:

$$\underset{\{\boldsymbol{\pi}, \boldsymbol{\theta}, \boldsymbol{m}, r\}}{\arg \min} E_{\{\boldsymbol{z}\}, \mathcal{D}}\left[L\left(\{\boldsymbol{z}, \boldsymbol{\theta}\}, \mathcal{D}\right)\right] \ \text{s.t.} \ \mathcal{E}\left(\{\gamma(\boldsymbol{\pi})\}\right) = e^*, \ \gamma(\boldsymbol{\pi}) = \text{onehot}\left(\underset{k}{\arg \max} \, \pi[k]\right) \tag{5}$$

where $L(\cdot)$ is a task loss, e.g. cross-entropy, $\mathcal{D}$ is the training data, and $e^*$ is the target model size. The constraint in (5) stems from the two-stage process typical in DNAS: 1) Optimize $\{\boldsymbol{\pi}, \boldsymbol{\theta}\}$ and extract the most likely configuration $\{\gamma(\boldsymbol{\pi})\}$, 2) Train and deploy the result [51, 21, 22, 40, 62, 65]. While prior works use the constraint [21, 22, 40, 62, 27]

$$E_{\{\boldsymbol{z}\}}\left[\mathcal{E}\left(\{\boldsymbol{z}\}\right)\right] = e^*, \tag{6}$$

we constrain the *most likely* configuration since this is what is actually deployed. Even if (6) is satisfied, it is not guaranteed that $\{\gamma(\boldsymbol{\pi})\}$, the deployed model, satisfies the constraint.

**Gradient-based optimization** Solving (5) using stochastic gradient descent (SGD) requires: 1) ensuring that the constraint is met, 2) differentiating with respect to $\{\boldsymbol{\theta}, \boldsymbol{\pi}\}$. To deal with the constraint and avoid dealing with the non-differentiable $\gamma(\boldsymbol{\pi})$, we modify (5) to

$$\underset{\{\boldsymbol{\pi},\boldsymbol{\theta},\boldsymbol{m},r\}}{\arg\min} \underbrace{E_{\{\boldsymbol{z}\},\mathcal{D}}\left[L\left(\{\boldsymbol{z},\boldsymbol{\theta}\},\mathcal{D}\right)\right]}_{\mathcal{L}_{\text{task}}} + \lambda \underbrace{E_{\{\boldsymbol{z}\}}\left[|\mathcal{E}\left(\{\boldsymbol{z}\}\right)-e^*|\right]}_{\mathcal{L}_{\mathcal{E}}^z}. \tag{7}$$

While $\mathcal{L}_{\mathcal{E}}^z$ has a different form than the constraint in (5), it actually represents a much stronger constraint and its minimization implies the constraint is met.

**Lemma 4.1.** *If $\mathcal{L}_{\mathcal{E}}^z = 0$, then for any sample of $\{\boldsymbol{z}\}$, denoted $\{\boldsymbol{z}^s\}$, with non-zero probability, $\mathcal{E}\left(\{\boldsymbol{z}^s\}\right) = e^*$ and $\mathcal{E}\left(\{\gamma(\boldsymbol{\pi})\}\right) = e^*$.*

$\mathcal{L}_{\mathcal{E}}^z$ differs from (6) because it penalizes any configuration which violates the constraint, whereas (6) penalizes $\{\boldsymbol{\pi}\}$ only if the expected storage size violates the constraint.

The derivative of (7) with respect to (w.r.t.) $\{\boldsymbol{\theta}\}$ can be approximated using a Monte-Carlo (MC) approximation of the expectation and applying standard automatic differentiation. The derivative w.r.t. $\{\boldsymbol{\pi}\}$ is more complex since the expectations in (7) depend on $\{\boldsymbol{\pi}\}$. A popular solution is to use a biased but differentiable approximation of $\boldsymbol{z}$, given by the Gumbel-softmax distribution: $\hat{\boldsymbol{z}} = \text{softmax}\left(\frac{\log\boldsymbol{\pi}+\boldsymbol{g}}{\tau}\right), g[k] \sim \text{Gumbel}\left(0,1\right), \tau > 0$ [39, 21, 22]. As $\tau \to 0$, $\hat{\boldsymbol{z}}$ approaches $\boldsymbol{z}$ in distribution, while the variance of any gradient estimator which uses $\hat{\boldsymbol{z}}$ increases ([55] and Fig. 5d). As a result, the common practice is to anneal $\tau$ from a high to a low value throughout the search. Using $\hat{\boldsymbol{z}}$ can lead to co-adaptation of search space options, which is undesirable but solvable in practice (Appendix D).

**Gumbel-softmax and over-regularization** We observe two issues with annealing $\tau$: 1) increased gradient variance at low $\tau$, coupled with a complex search space, causes issues for SGD, 2) when we replace $\boldsymbol{z}$ with $\hat{\boldsymbol{z}}$ in $\mathcal{L}_{\mathcal{E}}^z$, with the result $\mathcal{L}_{\mathcal{E}}^{\hat{z}}$, the regularizer becomes artificially inflated. To understand the impact of $\tau$ on $\mathcal{L}_{\mathcal{E}}^{\hat{z}}$, we evaluate $\mathcal{L}_{\mathcal{E}}^{\hat{z}}$ for different values of $\tau$, setting $\boldsymbol{\pi}$ such that $\mathcal{E}\left(\{\gamma(\boldsymbol{\pi})\}\right) = e^*$. The results

Table 2: Values of $\mathcal{L}_{\mathcal{E}}^{\hat{z}}$ for different values of $\tau, \xi$(see (8)), $\vartheta$(Sec. 4.2). The search space is based on MBNetV2 (Sec. 6) and $\mathcal{E}\left(\{\gamma(\boldsymbol{\pi})\}\right) = e^*$ in all cases.

| | Vanilla DNAS | Projection $\xi = 0.5, \vartheta = 0$ | Projection & Rejection Sampling $\xi = 0.5, \vartheta = 0.5$ | $\xi = 0.5, \vartheta = 0.99$ |
|---|---|---|---|---|
| $\tau = 0.66$ | 0.04 | 0.33 | 0.27 | 0.18 |
| $\tau = 10$ | 0.53 | 0.61 | 0.6 | 0.59 |

are presented in Table 2, col. 1 and show that increasing $\tau$ increases $\mathcal{L}_{\mathcal{E}}^{\hat{z}}$, i.e. the relative impact of $\mathcal{L}_{\mathcal{E}}^{\hat{z}}$ on (7) depends on $\tau$. As such, we seek to keep $\tau$ low, while minimizing gradient variance. Our solution is to use multiple samples of $\{\hat{\boldsymbol{z}}\}$ in the MC approximation of (7). To maintain the same computational cost as the single MC sample case, we divide the number of optimization steps by the number of samples. As well as reducing gradient variance, our strategy has two additional practical benefits: 1) trivial extension to multi-GPU systems, since each GPU can run its own MC sample and gradient computation, 2) the overheads of computing the gradient are amortized across the MC samples. We observe a 5.3× speed-up when going from 1 MC sample to 32 (Table 3, col. 6, row 4 vs. row 8.)

## 4.1 Exploration-exploitation

When solving (7), the goal is to explore as many configurations as possible (exploration), while still training each configuration for a meaningful number of steps (exploitation). Ideally, the search algorithm should gradually move from exploration to exploitation. We propose to explicitly control the exploration-exploitation trade-off by projecting $\boldsymbol{\pi}$ onto the set

$$\mathcal{S} = \left\{\boldsymbol{\pi} : \max_k \pi[k] \leq 1/|\boldsymbol{\pi}| + \xi^t\right\} \tag{8}$$

after each SGD step, where $\xi^t$ is the upperbound on $\boldsymbol{\pi}$ at step $t$. Setting $\xi^t = 0$ constrains $\boldsymbol{\pi}$ to parameterize a uniform distribution and represents maximal exploration.

Setting $\xi^t = 1 - 1/|\boldsymbol{\pi}|$ removes the constraint on $\boldsymbol{\pi}$, allowing the optimizer to enter full exploitation. We define the projection operator as $P_{\mathcal{S}}(\boldsymbol{\pi}) = \mathrm{softmax}\,(\log \boldsymbol{\pi}/T^*)$, $T^* = \arg\min_T \sum_k^{|\boldsymbol{\pi}|} \max\,(P_{\mathcal{S}}(\boldsymbol{\pi})[k] - (1/|\boldsymbol{\pi}| + \xi^t), 0)$, which we solve numerically. We choose this form for $P_{\mathcal{S}}(\boldsymbol{\pi})$ because of its simplicity and because the relative ordering of options between $\boldsymbol{\pi}$ and $P_{\mathcal{S}}(\boldsymbol{\pi})$ does not change. While we find it necessary to enforce exploration by projecting $\boldsymbol{\pi}$ onto $\mathcal{S}$, $\mathcal{L}_{\mathcal{E}}^z$ implicitly promotes exploitation.

**Lemma 4.2.** *Let $\mathcal{L}_{\mathcal{E}}^z = 0$ and $\boldsymbol{z}_j$ be the $j$'th decision variable. Let there be no decision for which two of its options have the same cost, i.e. for two configurations $\{\boldsymbol{z}^s\}$ and $\left\{\boldsymbol{z}^{s'}\right\}$ such that $\boldsymbol{z}_j^s \neq \boldsymbol{z}_j^{s'}$ and $\boldsymbol{z}_k^s = \boldsymbol{z}_k^{s'} \forall k \neq j$, we have $\mathcal{L}_{\mathcal{E}}^{z^s} \neq \mathcal{L}_{\mathcal{E}}^{z^{s'}}$. Then each $\boldsymbol{\pi}_j$ must be one-hot.*

The assumption in Lemma 4.2 that no decision has two options with the same storage cost is satisfied for a typical compressible model search space.

### 4.2 Combating over-regularization with rejection sampling

Projecting $\boldsymbol{\pi}$ onto $\mathcal{S}$ enables explicit control over the exploration-exploitation dynamics, but also inflates $\mathcal{L}_{\mathcal{E}}^{\hat{z}}$. Table 2, col. 2 shows that setting $\xi^t = 0.5$ increase $\mathcal{L}_{\mathcal{E}}^{\hat{z}}$ dramatically. By forcing $\boldsymbol{\pi}$ to be closer to uniform, the number of configurations $\{\hat{\boldsymbol{z}}\}$ with non-zero probability increases, such that the probability of a randomly drawn configuration violating the constraint also increases. Inflating $\mathcal{L}_{\mathcal{E}}^{\hat{z}}$ forces the optimizer to focus less on $\mathcal{L}_{\text{task}}$, leading to solutions which meet the constraint but perform poorly on the target task. $\mathcal{L}_{\mathcal{E}}^{\hat{z}}$ increases when the properties of $\{\boldsymbol{\pi}\}$ change because it depends on all possible configurations $\{\hat{\boldsymbol{z}}\}$ instead of the most likely one, i.e. $\{\gamma(\boldsymbol{\pi})\}$. To motivate the remedy, observe that while not all samples of $\hat{\boldsymbol{z}}$, denoted $\hat{\boldsymbol{z}}^s$, satisfy $\gamma(\hat{\boldsymbol{z}}^s) = \gamma(\boldsymbol{\pi})$, some do. Indeed, $\mathcal{L}_{\mathcal{E}}^{\hat{z}}$ would be 0 if it was evaluated over those samples that satisfy $\gamma(\hat{\boldsymbol{z}}^s) = \gamma(\boldsymbol{\pi})$, assuming $\hat{\mathcal{E}}\,(\{\gamma(\boldsymbol{\pi})\}) = e^*$ (and $\kappa = 1$ in Appendix D (12)). We refer to samples generated in this manner as $\tilde{\boldsymbol{z}}^s$ and they correspond to a RV whose distribution is different from $\hat{\boldsymbol{z}}$, but still depends on $\boldsymbol{\pi}$ and can therefore be used to generate gradients to $\boldsymbol{\pi}$ from $\mathcal{L}_{\mathcal{E}}^{\tilde{z}}$. Alg. 1 shows how to generate $\tilde{\boldsymbol{z}}^s$. Replacing $\hat{\boldsymbol{z}}$ with $\tilde{\boldsymbol{z}}$ for all decisions negates the effects of controlling $\boldsymbol{\pi}$ through projection onto $\mathcal{S}$. Therefore, we use $\tilde{\boldsymbol{z}}$ for a given decision with probability $\vartheta$ and $\hat{\boldsymbol{z}}$ otherwise.

---

**Algorithm 1** Rejection Sampling

1: Sample $\hat{\boldsymbol{z}}^s \sim p\,(\hat{\boldsymbol{z}})$, $1 \leq s \leq S$
2: $\tilde{s} = 1$, $k^* = \arg\max_k \boldsymbol{\pi}[k]$
3: **for** $1 \leq s \leq S$ **do**
4: $\quad \hat{k} = \arg\max_k \hat{\boldsymbol{z}}^s[k]$
5: $\quad$ **if** $\hat{k} = k^*$ **then**
6: $\quad\quad \tilde{\boldsymbol{z}}^{\tilde{s}} = \hat{\boldsymbol{z}}^s$, $\tilde{s} = \tilde{s} + 1$
7: $\tilde{z} = \frac{1}{\tilde{s}-1} \sum_{s'=1}^{\tilde{s}-1} \tilde{\boldsymbol{z}}^{s'}$ $\quad\quad$ ▷ Average
8: **return** $\tilde{z}$

---

**Algorithm 2** Complete `UDC` Algorithm

1: **for** $1 \leq t \leq t_{\max}$ **do**
2: $\quad$ **for** $1 \leq s \leq S$ **do**
3: $\quad\quad$ Generate MC sample $s$ for decision $j$ using $\hat{\boldsymbol{z}}_j$ w.p. $\vartheta$ and $\tilde{\boldsymbol{z}}_j$ else
4: $\quad\quad$ Update $\{\boldsymbol{m}\}$ if $t$ modulo $16 = 0$
5: $\quad\quad$ Take SGD step on (7)
6: $\quad\quad \boldsymbol{\pi} \leftarrow P_{\mathcal{S}}(\boldsymbol{\pi})$ $\quad\quad$ ▷ Projection
7: **return** $\{\boldsymbol{\pi}\}$

---

Using $\tilde{\boldsymbol{z}}$ has two major benefits. First, even when exploration is enforced by projecting $\boldsymbol{\pi}$ onto $\mathcal{S}$, $\mathcal{L}_{\mathcal{E}}^{\tilde{z}}$ is not inflated. Table 2, col.s 3-4 show how $\vartheta > 0$ brings down $\mathcal{L}_{\mathcal{E}}^{\hat{z}}$ for the same underlying $\{\boldsymbol{\pi}\}$. Second, mixing $\hat{\boldsymbol{z}}$ and $\tilde{\boldsymbol{z}}$ gives the flexibility of being in exploration for some decisions and exploitation for others. By randomly choosing which decisions use $\hat{\boldsymbol{z}}$ and which use $\tilde{\boldsymbol{z}}$, we prevent greedy behavior whereby a given decision enters exploitation and never returns to exploration. The complete `UDC` algorithm is summarized in Alg. 2.

## 5 Training sparse, quantized models

DNAS is typically a two-stage process: a search to find the model architecture is followed by finetuning to find optimal weights. However, we find that the second stage yields poor results when training sparse, quantized models from scratch [71]. `UDC` employs a three-stage finetuning process: **Stage 1:** initialize $\{\boldsymbol{\theta}\}$ and train with quantization enabled but unstructured pruning disabled, **Stage 2:** enable unstructured pruning gradually, **Stage 3:** train with both quantization and unstructured pruning enabled. To counter the training

challenges induced by quantizing and pruning $\boldsymbol{\theta}$, we employ several known techniques, with one slight modification. For quantization, we use a variant of [59] where each weight is quantized with probability $\alpha$ during the forward pass, such that the weights used for training are $Q(\boldsymbol{\theta}, b, r) \odot \boldsymbol{h} + \text{clip}(\boldsymbol{\theta}, -r, r) \odot (\mathbf{1} - \boldsymbol{h}), h[i, j, c_{in}, c_{out}] \sim \text{Bernoulli}(\alpha)$. Unlike UDC, [59] uses $\boldsymbol{\theta}$ in the second term. We observe that the range of $\boldsymbol{\theta}$ can differ significantly from that of $Q(\boldsymbol{\theta}, b, r)$, especially when unstructured pruning is applied during the learning process, so we clip $\boldsymbol{\theta}$ to the same range as $Q(\boldsymbol{\theta}, b, r)$. For unstructured pruning, we gradually anneal the pruning rate from 0% to 100% of the target rate during stage 2 [72, 50].

**Weight numerical representation** To understand the challenge of training sparse, quantized NNs, consider a single layer of pretrained weights, with histogram in Fig. 3a. The quantization bins ($b = 4$) are shown in red and the pruning boundary in purple, with everything between the purple lines mapped to 0. The sparse quantization problem is clearly apparent here: 8/14 of the non-zero quantization bins are unused because they fall *inside* the pruning boundary. If we now train with weight sparsity and quantization constraints, the weight distribution, quantization bins, and pruning zone adjust (Fig. 3b). The optimizer flattens the weight distribution to use more quantization bins, resulting in increased weight range. To quantify the weight growth, we report the norm of the NN weights before and after training in Fig. 3, showing an increase of over $4.6\times$. The rate at which NNs can be trained, known as the effective learning rate, is inversely proportional to the weight norm [12]. Therefore, the interaction of sparsity and quantization cause weight norm inflation, which decreases the effective learning rate, reducing NN performance. We propose a different weight representation and only quantize the range beyond the pruning boundary (Fig. 3c), using the quantization operator $\hat{Q}(\boldsymbol{\theta}, b, r, \beta) = Q(\boldsymbol{\theta} - \text{sign}(\boldsymbol{\theta})\beta, b, r) + \text{sign}(\boldsymbol{\theta})\beta$ for $b > 1, \beta \in \mathbb{R}$. We set $\beta$ to be the largest pruned value of $\boldsymbol{\theta}$. Training the sparse, quantized NN with the proposed number representation leads to much smaller weight norm (only $2.2\times$ growth over the pretrained weights), which makes training easier and accuracy higher (Fig. 5c).

**Deployment with integer math** NPU/MCU HW platforms typically only support integer operations, which are cheaper than floating point. There are at least two ways of deploying NNs quantized using the proposed approach on such HW. Firstly, convolution can be decomposed into $f(\boldsymbol{x}, Q(\boldsymbol{\theta} - \text{sign}(\boldsymbol{\theta})\beta, b, r)) + \beta f(\boldsymbol{x}, \text{sign}(\boldsymbol{\theta}))$. Both terms can be calculated using only integers, but since $\text{sign}(\boldsymbol{\theta})$ is a 1-bit tensor, the second term does not require any multiplications. Secondly, the NN can be trained using the proposed number format, then deployed with $Q\left(\hat{Q}(\boldsymbol{\theta}, b, r, \beta), b^*, r'\right)$, such that the deployed NN is uniformly quantized, with $b^* = 8$ to match the datatype supported in MCUs/NPUs [7, 8, 2]. The advantage of the latter approach is that the training benefits from the expressivity of the proposed number format, while its deployment uses a standard data type. We find that this approach with $b^* = 8$ does not incur an accuracy loss on ImageNet (Fig. 4a).

## 6 Results

We compare UDC with SOTA methods on a model size vs. accuracy basis. All reported model sizes, including related works, use compressed size (RHS of (4)), except Choi et al. [18], who use bzip2 to compress weights and we use their reported sizes.

**CIFAR100** is an image classification task with 50k training / 10k test images, and 100 classes. Our search space is based on the wide residual network with depth 20 and width multiplier 10 [70, 52]. We search over layer width (increments of 10% of the orignal), bitwidth (1, 4, 8, 32), and sparsity ($1 - 100\%$ non-zeros, increments of 10%). We use UDC ( settings in Appendix E) to find and train models at two sizes: 0.55MB and 0.7MB. Fig. 4b shows that UDC generates Pareto-dominant models compared with SOTA methods [27, 52, 65, 63, 58].

**ImageNet** is an image classification task with 1.28M training / 50k test images, and 1k classes. Our search space is based on MBNetV2, with the same options as for CIFAR100, and we target 0.5, 1, and 1.25 MB models (Appendix F). For the 1.25MB experiment, we replace $3\times3$ kernels with $5\times5$ and make all layers $1.5\times$ wider in the baseline architecture.

In practice, the model size constraints are determined by the Flash memory size of the deployment HW platform. To be sure, $0.5 - 1.25$ MB Flash sizes are fairly common for

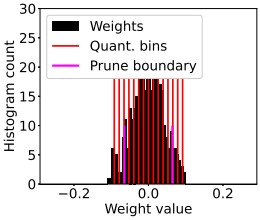 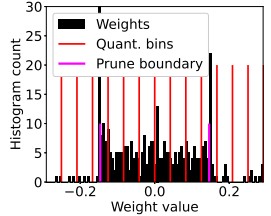 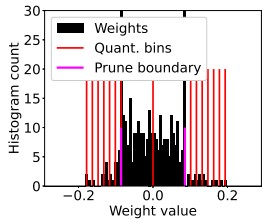

(a) Baseline, no retraining     (b) Baseline, retrained     (c) Proposed, retrained

Figure 3: Histogram of weights from a single pruned and quantized layer in the ImageNet experiment under different conditions. Weight norm $\|\theta\|_2^2$: (3a) 1.7e3, (3b) 7.9e3, (3c) 3.7e3.

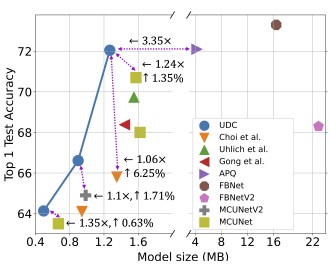 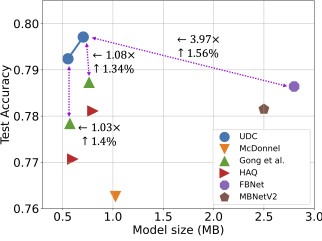 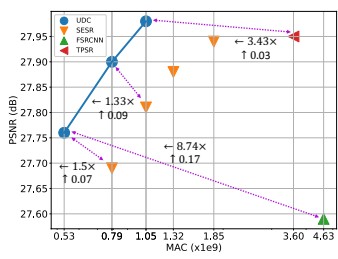

(a) ImageNet classification     (b) CIFAR100 classification     (c) set14 super resolution

Figure 4: `UDC` finds pareto-dominant models on image tasks.

commodity HW platforms [6] and are often used in research targeting deployment on constrained HW platforms [48, 14]. As such, we targeted this range because it represents a reasonable, but extremely challenging deployment scenario.

Fig. 4a shows that `UDC` generates pareto-dominant NNs vs. the SOTA [48, 18, 27, 49, 65, 62, 64]. Next, we include depth/operators in the search space, considering two alternatives to the inverted bottleneck: 1) regular 3×3 convolution, 2) identity. We target a 0.5MB model. `UDC` chooses inverted bottleneck blocks everywhere, such that the model found by including depth/operators in the search is identical to the one found when they are excluded. We compare `UDC` with non-uniform quantization approaches (Appendix H), and find that `UDC` models are pareto-dominant even though approaches like HAQ [63] and [66] employ a more expressive, non-uniform quantization which cannot be deployed on commodity MCUs and NPUs with integer math operations. Finally, comparing `UDC` to a SOTA unstructured pruning algorithm (Appendix I) shows that `UDC` finds considerably more accurate models.

**SR** The purpose of the SR experiment is to: 1) show that `UDC` can be applied to regression problems, 2) demonstrate `UDC` in a setup which constrains the computational complexity of the NN, measured in number of multiply-and-accumulate (MAC) operations, instead of model size. Note that 1 MAC = 2 floating point operations (FLOPs), but authors often conflate the two terms. This experiment excludes bitwidth and sparsity from the search space, which do not impact the inference compute cost. The topology of our search space is inspired by FSRCNN [20], with search over depth, width, and kernel size. We do all search/training on div2k and report results on set14 (see Appendix G for results on div2k and set5). We target 4x upscaling and report MACs for an input patch of 64x64. We train all NNs using the same settings. Fig. 4c shows that `UDC` finds Pareto-dominant NNs compared with SOTA efficient SR methods SESR, FSRCNN, and TPSR [15, 20, 44].

## 6.1 Comparison with random search

Random search is a strong baseline that can compete with DNAS in some settings [46]. We are not able to add early stopping [46], since sparsity is only applied during stages 2-3, when it is too late to save time from stopping. Fig. 5a compares `UDC` to randomly generated

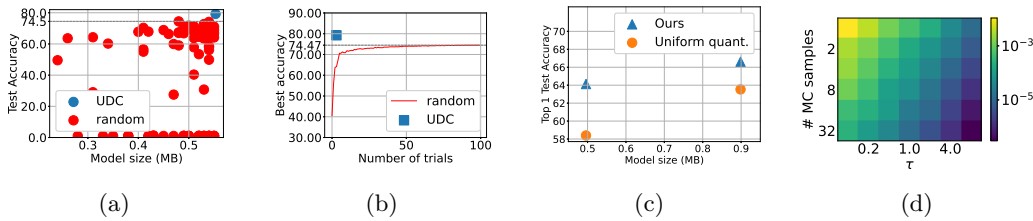

Figure 5: 5a-5b: Models found by random search. 5c: Benefit of novel number format on ImageNet. 5d: Maximum gradient variance over elements of $\{\boldsymbol{\pi}\}$ for CIFAR100 experiment.

Table 3: Ablation on `UDC` components for experiment targeting 0.5MB models. ∗–Constraint not met, so accuracy not reported. †–GPUD not reported as hyperparameter search for $\lambda$ is difficult to account for. For ImageNet, GPUD is normalized to the typical training cost of MBNetV2 (460 images / s) [4]) to allow for comparison to other works. Dashes indicate the experiment is prohibitively expensive to run because it requires hyperparameter search for $\lambda$.

| | Rejection Sampling | Multiple MC Samples | $\pi-$ Projection | No $\lambda$ Tuning | CIFAR100 Acc. (%) | GPUD | ImageNet Acc. (%) | GPUD |
|---|---|---|---|---|---|---|---|---|
| Vanilla DNAS | | | | | ∗ | † | — | — |
| Partial `UDC` | ✓ | ✓ | | ✓ | ∗ | † | ∗ | 3.2 |
| Partial `UDC` | | ✓ | ✓ | ✓ | 79.14 | 1.17 | ∗ | 3.2 |
| Partial `UDC` | | | ✓ | ✓ | 79.06 | 6.21 | — | — |
| Partial `UDC` | | ✓ | | ✓ | ∗ | **0.83** | ∗ | 3.2 |
| Partial `UDC` | ✓ | ✓ | | | 76.48 | † | — | — |
| Partial `UDC` | | ✓ | | | ∗ | † | — | — |
| **Full `UDC`** | ✓ | ✓ | ✓ | ✓ | **79.24** | 1.17 | **64.09** | **3.2** |

NNs constrained to model sizes $< 0.55$MB for CIFAR100, with the search space used in Fig. 4b. The gap between the best `UDC` models and random search is 4.77%. The best accuracy achieved by random search after a given number of trials is shown in Fig 5b. To integrate out randomness from the the ordering of trials, we permute the order of the random search results 100×, averaging over the permutations. Since `UDC` search requires ∼2× longer to find a NN than training a baseline NN, we plot the cost of `UDC` as 3 trials (2 to search and 1 to finetune). These results confirm that `UDC` finds better NNs faster than random search.

DNAS alternatives like evolutionary search require training many NNs or proxies to relax the compute burden [24, 23, 17, 48, 49]. However, using proxies assumes a search space dense with performant NNs, which is not true in our case (Fig. 5a).

## 6.2 Validation of algorithm and system components

**Compression** We verified our model size approximation, i.e. the RHS of (4) with pruning mask compressed to the entropy limit, by compressing $\boldsymbol{m}$ using arithmetic coding [53] for all `UDC` ImageNet models (Fig. 4a). We observed that theoretical and practical compressed model sizes are within two decimal places (Table 4). Table 4 shows that Vela can achieve com-

Table 4: Compressed model size (MB). Vela size represents Flash usage when deployed to NPU. MBNetV2 listed for reference.

| | Original | RHS of (4) | RHS of (4) & Compressed $m$ | Vela (Ratio) |
|---|---|---|---|---|
| Model 1 | 2.32 | 0.49 | 0.49 | 0.44 (5.27×) |
| Model 2 | 2.22 | 0.9 | 0.9 | 0.83 (2.67×) |
| Model 3 | 5.23 | 1.27 | 1.27 | 1.15 (4.55×) |
| MBNetV2 | 3.31 | 3.31 | 3.31 | 2.87 (1.15×) |

pression ratios even larger than those predicted by `UDC` for NPU deployment, which stems from the fact that non-zero weights can often be compressed to less than $b$ bits per weight by Golomb-Rice compression.

**Ablations** Table 3 gives an ablation on `UDC` components (Sec. 4) on CIFAR100 and ImageNet, which we evaluate on final accuracy, hyperparameter search over $\lambda$ requirement, and runtime in GPU days (GPUD) normalized to a typical training setup. Table 3 shows that disabling

Table 5: Algorithm runtime comparison on ImageNet. System is defined as software implementation and HW platform. Algorithms which begin with a pretrained model are assumed to pretrain the model for 200 epochs [4] and are marked by †.

| | Images / s (system & alg. specific) | Search epochs (alg. specific) | GPUD (system and alg. specific) | GPUD normalized to 460 im. / s (alg. specific) |
|---|---|---|---|---|
| UDC | 230 | 100 | 6.4 | 3.2 |
| FBNet [65] | 14.8 (11.5e6 images over 216 hours) | 90 (on 1/10 of ImageNet classes) | 9 | 0.3 |
| FBNetV2 [62] | 14.8 (11.5e6 images over 216 hours) | 90 (on 1/10 of ImageNet classes) | 9 | 0.3 |
| MCUNet [48] | — | 450 | — | 14.5 |
| MCUNetV2 [49] | — | 450 | — | 14.5 |
| Choi et al. [18] | — | $212.5^\dagger$ | — | 6.8 |
| Uhlich et al. [61] | — | $250^\dagger$ | — | 8 |

any of the UDC components leads to significantly worse results on both datasets. Vanilla DNAS [21] fails to meet the HW constraint. Fig. 5c shows the isolated benefit of the proposed number format over the baseline (3).

**Runtime** An important aspect in evaluating NAS algorithms is their runtime, or the time required to yield results. Part of the challenge in comparing algorithm runtimes is that the runtime generally depends on: 1) software implementation quality and HW platform, which jointly form the "system" that the algorithm runs on, 2) the algorithm itself, i.e. what the algorithm is actually doing during the search, 3) the number of epochs (or amount of data processed) for which the search is run. The challenge is that only 2-3 are algorithm dependent, whereas 1) depends on the code quality and the resources of the experimenter (i.e. higher grade GPUs exhibit higher throughput compared to low-end GPUs).

In order to compare with other works, we now attempt to disentangle the 3 components that make up the runtime in Table 5. We list the system and algorithm specific search speed measured in images per second, when available from the reference, the number of search epochs, the search cost in GPUD under the system used in the reference, and the search cost in GPUD under a common system assumed to be running at 460 images / s [4]. For several references, the algorithm begins with a pretrained model, which we assume is trained for a standard 200 epochs [4]. Table 5 shows that in absolute terms (i.e. the system specific cost), UDC is 1.4× faster than FBNet and FBNetV2. When comparing approaches based on a normalized system running at 460 images / s, UDC is faster than all of the competing approaches other than FBNet and FBNetV2. Our hope is that Table 5 gives a rough sense of the relative search cost of UDC and the competing methods.

**Societal impact & limitations** Developing UDC used hundreds of energy-consuming GPU hours. However, this can be amortized by increasing the energy efficiency of billions of IoT devices. A limitation of UDC is that we retrain $\{\boldsymbol{\theta}\}$ for every constraint $e^*$, whereas approaches based on OFA can amortize the cost of training $\{\boldsymbol{\theta}\}$ across multiple constraints.

## 7   Conclusion

Emerging NPU HW platforms specialized for TinyML support model compression, whereby quantized and pruned NNs can be stored in a reduced memory footprint. While compression is highly desirable, it increases the complexity of the NN design process, as the space of candidate NNs is increased by adding quantization and pruning on top of the conventional NN architecture choices. To enable TinyML practitioners to fully exploit HW model compression in NPUs, we describe a unified DNAS framework to search both architecture choices and aggressive per-layer quantization and pruning. We describe a number of improvements on top of DNAS, allowing us to demonstrate SOTA TinyML models that fully exploit model compression, as well as a comparison with random sampling and extensive ablations.

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
