## A    Proof of Lemma 4.1

*Proof.* We know that $p\left(\mathcal{E}\left(\{z\}\right) \neq e^*\right)$ must be 0. But if $\mathcal{E}\left(\{\gamma(\pi)\}\right) \neq e^*$, $p\left(\mathcal{E}\left(\{z\}\right) \neq e^*\right) > 0$, which is a contradiction. $\qquad\square$

## B    Proof of Lemma 4.2

*Proof.* If there is a $\pi_j$ which is not one-hot, then the following configuration sample has non-zero probability:

- $z_j^s \neq s_j^{s'}$

- $z_k^s = z_k^{s'} \forall k \neq j$

Since we assumed that $\mathcal{L}_\mathcal{E}\left(\{z\}\right) = 0$, Lemma 4.1 gives that $\mathcal{E}\left(\{z^s\}\right) = e^* = \mathcal{E}\left(\left\{z^{s'}\right\}\right)$. But this is a contradiction since we assumed there are no decisions for which two options have the same efficiency. $\qquad\square$

## C    Derivation of entropy bound

The entropy bound in (4) can be derived as follows:

$$H\left(Q(\boldsymbol{\theta}, b, r) \odot \boldsymbol{m}\right) \times \|\boldsymbol{w}\|_0 \tag{9}$$

$$\leq \left(H\left(Q(\boldsymbol{\theta}, b, r)\right) + H\left(\boldsymbol{m}\right)\right) \times \|\boldsymbol{w}\|_0 \tag{10}$$

$$\leq (b + \underbrace{H\left(\boldsymbol{m}\right)}_{-s\log_2 s - (1-s)\log_2(1-s)}) \times \|w\|_0 \tag{11}$$

where the first inequality follows from the fact that the entropy of a product of RVs is bounded by the sum of their entropies and the second bound follows from the fact that $Q(\boldsymbol{\theta}, b, r)$ costs at most $b$ bits per element to encode.

## D    Avoiding co-adaptation in DNAS

While samples of $\boldsymbol{z}$ are one-hot, samples of $\hat{\boldsymbol{z}}$ are not. This property can cause issues for an approach like DNAS, where weight-sharing can lead to co-adapation between search options [34, 29]. The result is a large performance drop when finetuning $\{\gamma(\boldsymbol{\pi})\}$, compared to the value of $\mathcal{L}_{\text{task}}$ achieved by solving (7) [36]. One solution, which we adopt, is to use a straight-through-estimator (STE), whereby

$$\hat{z}_{f_0}[k] = \begin{cases} \hat{z}[k] & k \in \text{topk}\left(\hat{\boldsymbol{z}}, \kappa\right) \\ 0 & \text{else} \end{cases} \quad | \quad \hat{\boldsymbol{z}}_f = \frac{\hat{z}_{f_0}}{\|\hat{z}_{f_0}\|_1} \tag{12}$$

is used in the forward pass, where $\kappa$ is the number of non-zeros in $\hat{z}_f[k]$, topk$(\hat{\boldsymbol{z}}, \kappa)$ returns the indices of the $\kappa$ largest elements of $\hat{\boldsymbol{z}}$, and $\hat{\boldsymbol{z}}$ is used in the backward pass [39]. Typically, $\kappa \in \{1, 2\}$.

## E    CIFAR100 experiment settings

We run the search for 200 epochs, annealing $\tau$ from 0.66 to 0.1 using an exponential schedule. We use SGD for $\boldsymbol{\theta}$ with learning rate annealed from 0.1 to $1e-4$ using a cosine schedule and we use ADAM for $\{\pi\}$ with a constant learning rate of $1e-3$. We increase $\vartheta$ from 0 to 0.5 using a linear schedule and we increase $\zeta^t$ from 0.1 to 1 using a linear schedule. We initialize the search by running a warmup stage for 50 epochs where we use SGD for $\boldsymbol{\theta}$ with learning rate annealed from 0.1 to $1e-4$ using a cosine schedule, $\tau$ is annealed from 0.66 to 0.1 using an exponential schedule, $\{\pi\}$ is not learned, and $\vartheta = 0$. During both warmup and search, we set $\kappa = K$ for all width decisions and $\kappa = 2$ for all quantization and sparsity decisions.

|              | Top1 acc. (%) | Model size (MB) |
|--------------|---------------|-----------------|
| UDC          | **79.24**     | **0.553**       |
| Gong et al. [27] | 77.84     | 0.57            |
| HAQ          | 77.07         | 0.6             |
| UDC          | **79.71**     | **0.705**       |
| Gong et al. [27] | 78.73     | 0.76            |
| McDonnel, [51] | 76.26       | 1.02            |
| HAQ          | 78.11         | 0.8             |
| FBNet        | 78.64         | 2.8             |
| MBNetV2      | 78.15         | 2.5             |

Table 6: Detailed CIFAR100 results comparing compressed model size versus accuracy for UDC against SOTA algorithms. Note that HAQ uses non-uniform quantization, such that models produced by HAQ cannot be deployed on commercial NPUs running integer convolutions.

To finetune the discovered models, we run stage 1 for 254 epochs using SGD and cosine decay with restarts learning rate schedule, cycling between 0.1 and $1e-4$ at intervals which double after every cycle and beginning with a cycle of 2 epochs. We run stage 2 for 60 epochs, annealing the learning rate from 0.1 to $1e-4$ using a cosine schedule and then we run stage 3 for 30 epochs, annealing the learning rate from $1e-4$ to 0 using a cosine schedule. We use distillation with a teacher model whose architecture is WRN 20-10. For data augmentation, we use horizontal flipping, random crop with a size of 4, and cutout with a size of 18. We use $\ell_2$ regularization with coefficient $5e-4$. We disable learning of batchnorm scale and offset parameters [52].

Table 6 presents the detailed experimental results for the CIFAR100 experiments.

## F   ImageNet experiment settings

We use the same search settings as for the CIFAR100 experiments. To finetune the discovered models, we run stage 1 for 126 epochs using SGD and cosine decay with restarts learning rate schedule, cycling between 0.1 and $1e-4$ at intervals which double after every cycle and beginning with a cycle of 2 epochs. We run stage 2 for 60 epochs, annealing the learning rate from 0.1 to $1e-4$ using a cosine schedule and then we run stage 3 for 30 epochs, annealing the learning rate from $1e-4$ to 0 using a cosine schedule. For the 0.5 and 1MB target experiments, we use distillation with a teacher model whose architecture is MobileNetV2. We do not use distillation for the 1.25MB target experiment. For data augmentation, we use the standard ImageNet data pipeline [22], as well as horizontal flipping and label smoothing with smoothing coefficient 0.1. We use $\ell_2$ regularization with a coefficient of $1e-4$.

Table 7 shows the detailed ImageNet results.

## G   Super resolution experiment settings

We run the search for 300 epochs, with constant $\tau$ set to 0.1. We use ADAM for $\boldsymbol{\theta}$ with learning rate annealed from $1.e-4$ to $1e-5$ using a cosine schedule and ADAM as well for $\{\pi\}$ with a constant learning rate of $1e-3$. We keep $\vartheta$ constant to 0.25 and we increase $\zeta^t$ from 0.1 to 1 using a cosine schedule.

Table 8 provides detailed results for the super resolution experiment.

## H   Comparison to non-uniform quantization approaches

We compare UDC to approaches which employ non-uniform quantization in Fig. 7. UDC is Pareto-dominant even though it uses uniform quantization and can be deployed on MCUs/NPUs with integer math whereas the other approaches cannot.

|  | Top1 acc. (%) | Model size (MB) |
|---|---|---|
| **UDC** | **64.13** | **0.5** |
| MCUNet | 63.5 | 0.67 |
| **UDC** | **66.61** | **0.9** |
| MCUNetV2 | 64.9 | 0.99 |
| Choi et al., [18] | 64.1 | 0.94 |
| MCUNetV2 | 64.9 | 0.99 |
| **UDC** | **72.05** | **1.27** |
| Choi et al., [18] | 65.8 | 1.35 |
| MCUNet | 70.7 | 1.57 |
| Gong et al., [27] | 68.38 | 1.44 |
| Uhlich et al., [60] | 69.74 | 1.55 |
| APQ | 72.1 | 4.26 |
| FBNet | 73.3 | 16.4 |
| FBNetV2 | 68.3 | 22.89 |

Table 7: Detailed ImageNet experimental results, comparing compressed model size versus accuracy for `UDC` and SOTA algorithms.

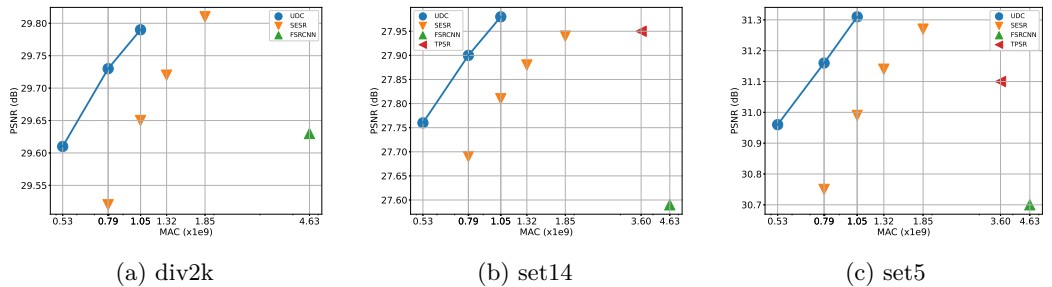

(a) div2k      (b) set14      (c) set5

Figure 6: Super resolution results for $4\times$ upsampling. MACs are reported for an input size of $64 \times 64$.

|  | div2k | | set14 | | set5 | |
|---|---|---|---|---|---|---|
|  | PSNR (dB) | MAC (x1e9) | PSNR (dB) | MAC (x1e9) | PSNR (dB) | MAC (x1e9) |
| UDC | **29.61** | **0.53** | **27.76** | **0.53** | **30.96** | **0.53** |
| SESR | 29.52 | 0.79 | 27.69 | 0.79 | 30.75 | 0.79 |
| UDC | **29.73** | **0.79** | **27.9** | **0.79** | **31.16** | **0.79** |
| SESR | 29.52 | **0.79** | 27.69 | **0.79** | 30.75 | **0.79** |
| UDC | **29.79** | **1.05** | **27.98** | **1.05** | **31.31** | **1.05** |
| SESR | 29.65 | **1.05** | 27.81 | **1.05** | 30.99 | **1.05** |
| SESR | 29.72 | **1.32** | 27.88 | **1.32** | 31.14 | **1.32** |
| SESR | **29.81** | 1.85 | **27.94** | 1.85 | **31.27** | 1.85 |
| FSRCNN | 29.63 | 4.63 | 27.59 | 4.63 | 30.7 | 4.63 |
| TPSR | — | — | 27.95 | 3.6 | 31.1 | 3.6 |

Table 8: Detailed super resolution experiment results comparing `UDC` to SOTA efficient super resolution algorithms. MACs are reported for 4x upsampling with an input of size $64 \times 64$.

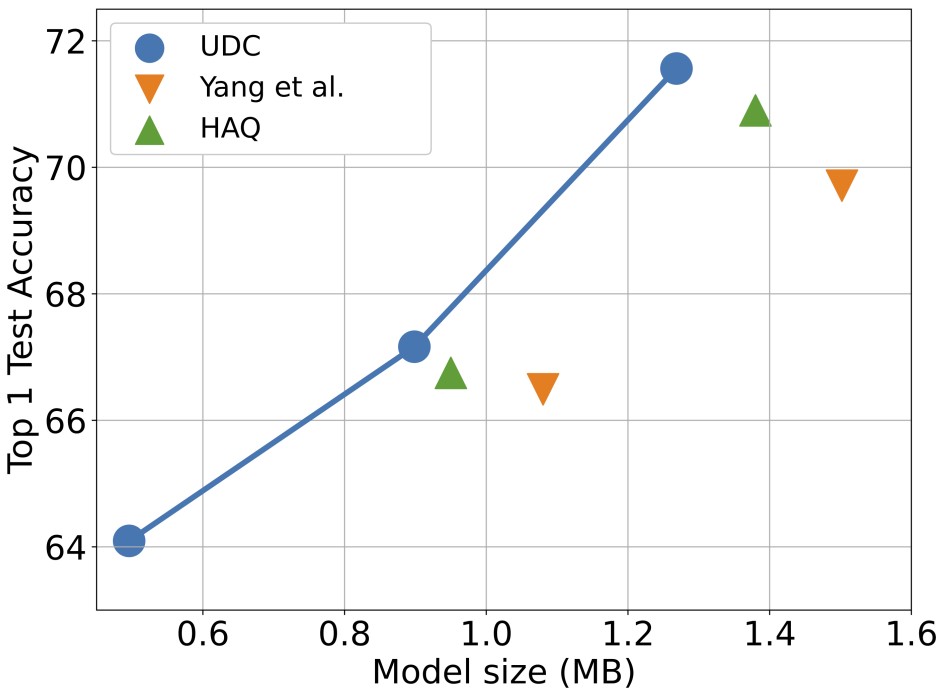

Figure 7: ImageNet test set accuracy vs. compressed model size.

# I   Comparison to approaches which only do unstructured pruning

We also compare `UDC` to a SOTA unstructured pruning algorithm [43] in Fig. 8. As the results show, `UDC` generates much more accurate models.

# J   Visualization of design choices

We provide a visualization of the design choices made by `UDC` for the ImageNet experiments in Fig. 9-11.

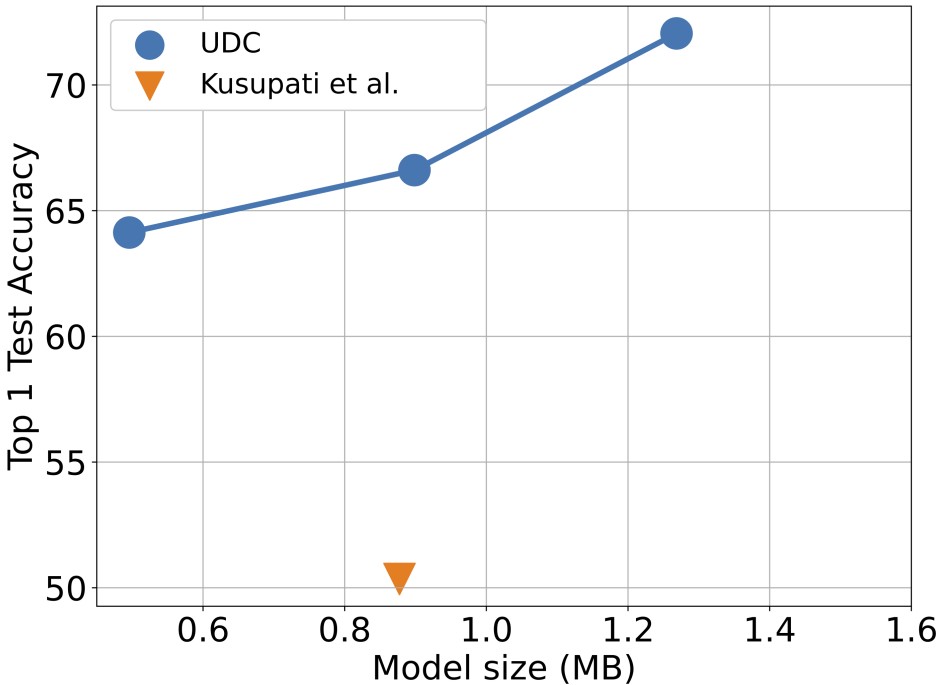

Figure 8: ImageNet test set accuracy vs. compressed model size.

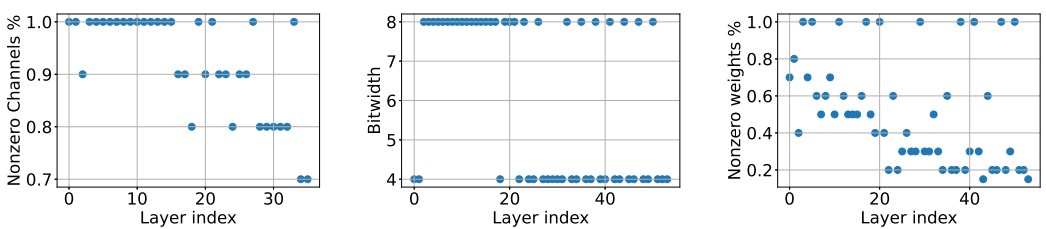

(a) Channel width decisions for 0.5 MB ImageNet experiment

(b) Quantization decisions for 0.5 MB ImageNet experiment

(c) Unstructured pruning for 0.5 MB ImageNet experiment

Figure 9: Model decisions for 0.5MB ImageNet experiment.

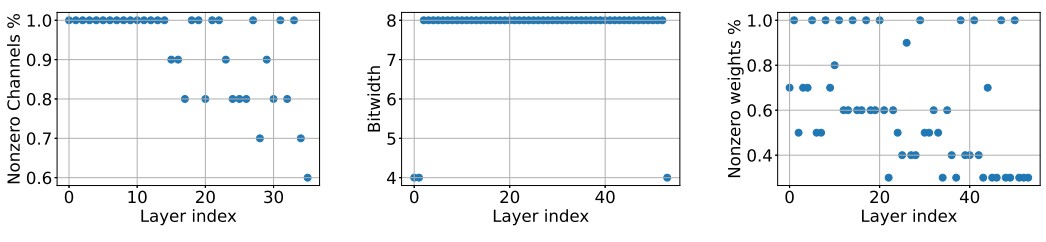

(a) Channel width decisions for 1 MB ImageNet experiment

(b) Quantization decisions for 1 MB ImageNet experiment

(c) Unstructured pruning decisions for 1 MB ImageNet experiment

Figure 10: Model decisions for 1 MB ImageNet experiment.

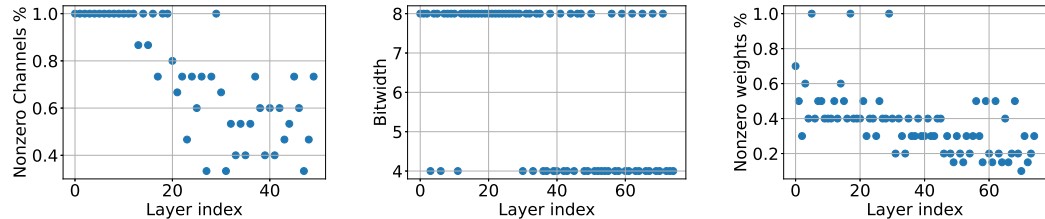

(a) Channel width decisions for 1.25 MB ImageNet experiment

(b) Quantization decisions for 1.25 MB ImageNet experiment

(c) Unstructured pruning decisions for 1.25 MB ImageNet experiment

Figure 11: Model decisions for 1.25 MB ImageNet experiment.