# OpenReview forum: "UDC: Unified DNAS for Compressible TinyML Models for Neural Processing Units"
_NeurIPS.cc/2022/Conference — NeurIPS 2022 Accept_

### Official Review · Reviewer_2bCy · 2022-07-07

**Rating:** 6
**Confidence:** 3
**Soundness:** 2 fair
**Presentation:** 2 fair
**Contribution:** 2 fair

**Summary:**

This paper is on model compression (quantize and prune) with hardware constraints. The proposed search method is an extension of differentiable NAS to learn weight sparsity ratios and bit-widths per layer. Conceptually, this work combines previous ideas, such as DNAS (of layer width, depth, or operator) and learning of layer-wise bitwidth and sparsity, and yield pareto-optimal results on model size vs. accuracy.


**Questions:**

- What is the role of the Vela compiler in the proposed framework? Are the results on model size, e.g., in Fig. 4 & Fig. 5, from the use of the Vela compiler afterwards?
- What are the performance results, in terms of latency/inference time, of the searched, compressed, and deployed models? Setting the optimization goal to be smallest model size is reasonable if the model size matters the most. But what is the impact on inference performance? The work mentioned using unstructured pruning, would that hurt the performance? Please comment on that.
- Can you compare the searching costs with other work? For example, the GPU days and training FLOPs could be helpful in understanding this work when compared with others.

**Limitations:**

The authors have addressed the potential societal impact.
The limitations are not addressed enough.

**Strengths And Weaknesses:**

- The methods used in the paper are not new. But this work is a sort of combination of well-known techniques. Thus, making it work, i.e., DNAS for model architecture with pruning and quantization, is a new contribution. The related work part discusses the differences from prior work, and the related work is adequately cited to my knowledge.
- The submission is technically sound with mostly empirical results. But it does not evaluate the weaknesses. For example, the searching costs of the proposed algorithm.
- The writing is okay. The design decisions and the proposed algorithm are described with details. Since the major contribution, to my understanding, is the practical issues of making DNAS with compression techniques working, I think more details are necessary to help readers and practitioners to use the findings in this work. For example,  experiment setups and code examples are helpful, especially if no source code available.
- The results are useful but not significant. In some scenarios with strict limitations on model size, the proposed method could be used. The reported benefits (model size vs. accuracy) are not unique.

---

> ### Author Response · Authors · 2022-07-30
> **Response to reviewer 2bCy (1/2)**
>
> We thank the reviewer for providing feedback on our work. In the following, we address the reviewer’s questions.
>
> > For example, experiment setups and code examples are helpful, especially if no source code available
>
> The detailed experimental setups are described in Appendices E-G. The complete algorithm pseudo-code is presented in Algorithms 1-2, line 220. We are happy to provide some additional pseudo-code in the revision, so as to make using our approach easier by others.
>
> > The results are useful but not significant. In some scenarios with strict limitations on model size, the proposed method could be used
>
> In the TinyML setting, where models are deployed on resource constrained embedded hardware, model size is a hugely important factor because of the severely limited Flash storage capacity (see the references in lines 26-28). Therefore, taking full advantage of the model compression ability of modern NPUs offers a unique opportunity for deploying more accurate models on more efficient hardware.
>
> > What is the role of the Vela compiler in the proposed framework? Are the results on model size, e.g., in Fig. 4 & Fig. 5, from the use of the Vela compiler afterwards?
>
> As we discuss in lines 272-274, we use Eq. (2) to measure compressed model size in all figures and for all competing methods, including UDC. Vela is not used in any of our results comparing UDC to existing NAS methods. We introduce the Vela compiler / compression engine as motivation for our work and as a real-world realization of a deployment scenario which uses model compression. The only place where we use Vela to report compressed model sizes is in Table 4, where we verify that the compression rate predicted by Eq. (2) is achievable by the Vela compiler (indeed Vela achieves even higher compression ratio compared to Eq. (2)).
>
> > What are the performance results, in terms of latency/inference time, of the searched, compressed, and deployed models? Setting the optimization goal to be smallest model size is reasonable if the model size matters the most. But what is the impact on inference performance? The work mentioned using unstructured pruning, would that hurt the performance? Please comment on that.”
>
> In the use case we studied, decompression is implemented in the Ethos U55 hardware. Since all models pass through a decompression stage during inference, there is no penalty for executing compressed models (i.e. as generated by UDC) compared to off-the-shelf models on the Ethos U55 hardware (or any hardware which has HW decompression). Moreover, for a given model, weight compression leads to memory bandwidth savings, further compounding the benefit at inference time. Reducing model size allows for larger portions of weight stationary execution through compiler optimizations such as cascading (https://developer.arm.com/documentation/101888/0500/NPU-software-overview/NPU-software-tooling/The-Vela-compiler). These optimizations result in less data movement from external memory, and thus, less energy consumption and possibly lower latency.
>
> As a side note, although our models are heavily pruned and therefore highly compressible by Vela, there is no computational benefit to this pruning since neither the hardware nor inference software stack are designed to take advantage of sparsity to save computation. As such, we don’t expect much of a latency improvement when comparing a given model to its pruned and quantized counterpart. With that being said, it is possible to design future TinyML hardware that takes advantage of model sparsity and more aggressive quantization to give good latency improvement as well.

---

> > ### Author Response · Authors · 2022-07-30
> > **Response to reviewer 2bCy (2/2)**
> >
> > > Can you compare the searching costs with other work? For example, the GPU days and training FLOPs could be helpful in understanding this work when compared with others
> >
> > We appreciate the reviewer’s comments about the relative runtime of UDC compared to other NAS algorithms. Part of the challenge in comparing algorithm runtimes is that the runtime generally depends on 3 things: 1) Software implementation quality and hardware platform, which jointly form the “system” that the algorithm runs on, 2) the algorithm itself, i.e. what the algorithm is actually doing during the search, 3) the number of epochs (or amount of data processed) for which the search is run. The challenge is that only 2-3 are algorithm dependent, whereas 1) depends on the code quality and the resources of the researcher (i.e. some researchers may be able to afford higher grade GPUs which exhibit much higher throughput compared to low-end GPUs).
> >
> > In the paper, we tried to address the runtime question by saying that UDC runs at roughly half the speed (230 images / s) of a typical training experiment (460 images / s) on ImageNet. Although we did not state this in the paper, we also observed that UDC’s speed is nearly independent of the search space size in the ImageNet experiment. In other words, introducing quantization and unstructured pruning into the search does not lead to a slowdown compared to only searching for width. Likewise, increasing the number of options (i.e. how many distinct sparsity rates to consider) also does not lead to a slowdown. This suggests that UDC’s search speed is highly scalable with the search space size and that most of the slowdown can be accounted for by the system which the algorithm is running on. For that reason, we list the search cost of UDC in the last column of Table 3 as 3.2 GPUD normalized to a system running at 460 images / s.
> >
> > In order to compare with other works, we now attempt to disentangle the 3 components that make up the search cost in the table below. We list the system and algorithm specific search speed, when available from the reference, the number of search epochs, the search cost in GPUD under the system used in the reference, and the search cost in GPUD under a common system assumed to be running at 460 images / s. For several references, the algorithm begins with a pretrained model, which we assume is trained for a standard 200 epochs. The table shows that in absolute terms (i.e. the system specific cost), UDC is 1.4x faster than FBNet / FBNetV2. When comparing approaches based on a normalized system running at 460 images / s, UDC is faster than all of the competing approaches other than FBNet / FBNetV2. Our hope is that the table below gives a rough sense of the relative search cost of UDC and the competing methods.
> >
> > | Algorithm | Images / s (system and algorithm specific) | search epochs (algorithm specific) | GPUD (system and algorithm specific) | GPUD normalized to 460 images / s (algorithm specific) |
> > | --- | --- | --- | --- | --- |
> > UDC |	230 | 100 | 6.4 | 3.2 |
> > FBNet |	14.8 (calculated using details in reference: 11.5 million images, processed over 216 GPU hours) |	90 (on 1/10 of ImageNet classes)	| 9	| 0.3 |
> > FBNetV2 |	14.8 (calculated using details in reference: 11.5 million images, processed over 216 GPU hours) |	90 (on 1/10 of ImageNet classes)	| 9 |	0.3 |
> > MCUNet	| -- |	450 |	--	| 14.5 |
> > MCUNetV2 |	-- |	450 |	--	| 14.5 |
> > Gong et al. |	-- |	-- |	-- |	-- |
> > Choi et al. |	-- |	200 + 12.5 |	--	| 6.8 |
> > Uhlich et al. |	-- |	200 + 50 |	-- |	8 |

---

### Official Review · Reviewer_c4gz · 2022-07-08

**Rating:** 6
**Confidence:** 3
**Soundness:** 3 good
**Presentation:** 4 excellent
**Contribution:** 4 excellent

**Summary:**

The authors present am updated Differential Neural Architecture Search algorithm, UDC, which incorporates model compression features such as weight sparsity and quantization as part of the search space for the ideal (high accuracy, small model size) neural network (NN) model. In addition, keeping in mind the constraints typically faced by TinyML models, the authors also enforce a strict limit on the model size to ensure it can be implemented in resource constrained IoT applications. Further, the presented algorithm enables effective exploration of accuracy vs model size tradeoff. Lastly, the paper demonstrates the pareto dominance of NNs designed by UDC compared to prior art for a range of ML benchmarks and varying hardware constraints on Ethos U55 NPU.

**Questions:**

Refer to questions addressed in Strengths/Weakness section.

**Limitations:**

Authors do not present any data showing the timing performance of UDC compared to other approaches, some perspective on that would be helpful. Also any explanation on how the memory size constraints were selected for presented experiments would be helpful.

**Strengths And Weaknesses:**

The paper presents a thorough comparison of UDC against past techniques that attempt similar co-optimization of ML models while accounting for hardware constraints. UDC tackles a wider range of constraints compared to all listed in prior art.

Section 3 and 4 describe the details of the design model and the proposed DNAS algorithm. These sections were somewhat hard to follow, however, it is understandable considering the mathematical nature of problem. It is not obvious what these variables denote, ξ & ν, kindly add a description.

Section 6 shows a comparative study of generating NNs using UDC and other prior art for state-of-the-art (SOTA) ML benchmarks under different memory size constraints. Figure 4 illustrates the superiority of UDC in terms of accuracy and model size, could the authors also shed light on the runtime it takes to converge to the final NN design UDC picks compared to other approaches (e.g., MCUNet). Considering UDC accounts for a larger set of constraints compared to other approaches it would be interesting if it can demonstrate comparable runtimes too.

Figure 5b, compares UDC results to random search, and the authors do share some insight on UDC runtime compared to training a baseline network (~2x), if they could share this runtime information it would be handy. Also note that authors should be careful that while comparing to random search the resource utilization for UDC should be similar e.g., running equal number of parallel GPU instances for a fair comparison. It would be interesting to know how long it takes to converge the random search to the optimal design picked up by UDC (withing reasonable time constraints of course).

Additionally, could authors share how they selected the memory size constraint values for the experiments shared in the paper, is there some understanding if the superiority of UDC holds for larger/smaller mem constraints and if yes how a user could understand if they fall in that range.

To conclude, this work promises a very useful optimization tool for designing small but high accuracy NNs that can fit on typical IoT devices servicing TinyML applications.

---

> ### Author Response · Authors · 2022-07-30
> **Response to reviewer c4gz (1/2)**
>
> We thank the reviewer for the thorough review and feedback. In the following, we address the reviewer’s concerns point by point.
>
> > It is not obvious what these variables denote, ξ & ν, kindly add a description.
>
> The symbols are already defined in the paper ($\xi$ is defined in line 191 and $\nu$ is defined in line 219), but we will make sure to highlight their meaning with further description in the revision.
>
> > Figure 4 illustrates the superiority of UDC in terms of accuracy and model size, could the authors also shed light on the runtime it takes to converge to the final NN design UDC picks compared to other approaches (e.g., MCUNet)
>
> We appreciate the reviewer’s comments about the relative runtime of UDC compared to other NAS algorithms. Part of the challenge in comparing algorithm runtimes is that the runtime generally depends on 3 things: 1) Software implementation quality and hardware platform, which jointly form the “system” that the algorithm runs on, 2) the algorithm itself, i.e. what the algorithm is actually doing during the search, 3) the number of epochs (or amount of data processed) for which the search is run. The challenge is that only 2-3 are algorithm dependent, whereas 1) depends on the code quality and the resources of the researcher (i.e. some researchers may be able to afford higher grade GPUs which exhibit much higher throughput compared to low-end GPUs).
>
> In the paper, we tried to address the runtime question by saying that UDC runs at roughly half the speed (230 images / s) of a typical training experiment (460 images / s) on ImageNet. Although we did not state this in the paper, we also observed that UDC’s speed is nearly independent of the search space size in the ImageNet experiment. In other words, introducing quantization and unstructured pruning into the search does not lead to a slowdown compared to only searching for width. Likewise, increasing the number of options (i.e. how many distinct sparsity rates to consider) also does not lead to a slowdown. This suggests that UDC’s search speed is highly scalable with the search space size and that most of the slowdown can be accounted for by the system which the algorithm is running on. For that reason, we list the search cost of UDC in the last column of Table 3 as 3.2 GPUD normalized to a system running at 460 images / s.
>
> In order to compare with other works, we now attempt to disentangle the 3 components that make up the search cost in the table below. We list the system and algorithm specific search speed, when available from the reference, the number of search epochs, the search cost in GPUD under the system used in the reference, and the search cost in GPUD under a common system assumed to be running at 460 images / s. For several references, the algorithm begins with a pretrained model, which we assume is trained for a standard 200 epochs. The table shows that in absolute terms (i.e. the system specific cost), UDC is 1.4x faster than FBNet / FBNetV2. When comparing approaches based on a normalized system running at 460 images / s, UDC is faster than all of the competing approaches other than FBNet / FBNetV2. Our hope is that the table below gives a rough sense of the relative search cost of UDC and the competing methods.
>
> | Algorithm      |  Images / s (system and algorithm specific) | search epochs (algorithm specific) | GPUD (system and algorithm specific) |  GPUD normalized to 460 images / s (algorithm specific) |
> | ----------- | ----------- | ----------- | ----------- | ----------- |
> | UDC      | 230       | 100 | 6.4 | 3.2 |
> | FBNet   | 14.8 (calculated using details in reference: 11.5 million images, processed over 216 GPU hours)        | 90 (on 1/10 of ImageNet classes) | 9 | 0.3 |
> | FBNetV2   | 14.8 (calculated using details in reference: 11.5 million images, processed over 216 GPU hours)        | 90 (on 1/10 of ImageNet classes) | 9 | 0.3 |
> MCUNet | -- | 450 | -- | 14.5 |
> MCUNetV2 | -- | 450 | -- | 14.5 |
> Gong et al. | -- | -- | -- | -- |
> Choi et al. | -- | 200 + 12.5 | -- | 6.8 |
> Uhlich et al. | -- | 200 + 50 | -- | 8
>
> > Also note that authors should be careful that while comparing to random search the resource utilization for UDC should be similar e.g., running equal number of parallel GPU instances for a fair comparison
>
> Indeed, this is a great point. Since we quantify search cost in terms of number of trials in Fig. 5b and each trial uses the same amount of GPU resources, the data does in fact provide a fair comparison between UDC and random search.

---

> > ### Author Response · Authors · 2022-07-30
> > **Response to reviewer c4gz (2/2)**
> >
> > > It would be interesting to know how long it takes to converge the random search to the optimal design picked up by UDC (within reasonable time constraints of course).
> >
> > We agree that it would be interesting to conduct this study, but we can already see that UDC achieves significantly better results compared to random search, using a small fraction of the trials. We did not have spare computational resources to extend the random search to multiple hundreds or thousands of trials, but we will try and generate these results for the revision.
> >
> > > Additionally, could authors share how they selected the memory size constraint values for the experiments shared in the paper
> >
> > In practice, the model size constraints are determined by the Flash memory size of the deployment hardware platform. To be sure, 0.5-1.25 MB Flash sizes are fairly common for commodity hardware platforms (i.e. STM32F446RE, STM32F746ZG, STM32F767ZI) and are often used in research papers targeting deployment on constrained hardware platforms [1,2]. As such we targeted this range because it represents a reasonable, but extremely challenging deployment scenario.
> >
> > > is there some understanding if the superiority of UDC holds for larger/smaller mem constraints and if yes how a user could understand if they fall in that range.
> >
> > Since we only experimented with UDC in the very low to medium model size regime, we are hesitant to speculate how well UDC would perform in the very high model size regime. With that being said, we can see in Fig. 4a that UDC finds an ImageNet model with >72% accuracy while being 3.35x smaller than an APQ model at 4MB, which is considered a rather large model size in the context of resource constrained hardware deployment. This suggests that UDC does scale well when moving from very small to medium sized models.
> >
> > [1] Ji Lin, Wei-Ming Chen, Yujun Lin, Chuang Gan, Song Han, et al. "Mcunet: Tiny deep learning on iot devices." Advances in Neural Information Processing Systems, 2020.
> >
> > [2] Colby Banbury, Chuteng Zhou, Igor Fedorov, Ramon Matas, Urmish Thakker, Dibakar Gope, Vijay Janapa Reddi, Matthew Mattina, and Paul Whatmough. "Micronets: Neural network architectures for deploying tinyml applications on commodity microcontrollers." Proceedings of Machine Learning and Systems, 3, 2021.

---

### Official Review · Reviewer_uYMT · 2022-07-11

**Rating:** 6
**Confidence:** 3
**Soundness:** 3 good
**Presentation:** 3 good
**Contribution:** 3 good

**Summary:**

This paper presents a differentiable NAS approach targetting compressible NN for NPUs with low memory footprint.
The approach conducts a joint search over NN architecture, weight bitwidths and layer-wise weight sparsity levels to find models with the smallest model size.

**Questions:**

- What is the decompression overhead when the compressed model is executed on the target device?

**Limitations:**

The limitations are adequately discussed.

**Strengths And Weaknesses:**

Strengths:
- A theoretical lower-bound of the storage size that is cheap to be computed.
- Clear three-stage training process to deal with quantization and sparsity.
- Improved pareto-front of model size vs accuracy over previous work.
- The search algorithm has better sample efficiency vs random method.

Weaknesses:
- Make it clear that the target device is mobile-based NPU with limited model storage.
- Models targetting small NPU and MCU have strigent latency and memory utilization constraints, which are not discussed in this paper.
- Limited comparison with other NAS algorithms.
- There are other NAS papers for MCU, please also cite and compare to them. For example: https://arxiv.org/abs/2010.14246

---

> ### Author Response · Authors · 2022-07-30
> **Response to Reviewer uYMT**
>
> We thank the reviewer for the feedback. In the following, we address the reviewer’s concerns point by point.
>
> > Make it clear that the target device is mobile-based NPU with limited model storage.
>
> We tried to be as clear as possible about this point. The abstract states our problem setup as deployment onto memory constrained NPUs (lines 1-2). We state that our target hardware platform is a Flash memory-limited NPU (line 26). We give the problem statement in (P1), which makes clear that our goal is to design small models (in terms of memory cost) while exploiting NPU model compression (line 32).
>
> > Models targeting small NPU and MCU have stringent latency and memory utilization constraints, which are not discussed in this paper.
>
> Our goal was to explore the intersection of model compression and NAS, since hardware weight decompression is a new feature of modern NPUs like Arm Ethos U55. Model compression can have a significant impact in NPUs like Ethos: reducing model size allows for larger portions of weight stationary execution through compiler optimizations such as cascading (https://developer.arm.com/documentation/101888/0500/NPU-software-overview/NPU-software-tooling/The-Vela-compiler). These optimizations result in less data movement from external memory, and thus, less energy consumption and possibly lower latency. With that being said, we completely agree that real-world use cases must also consider latency and SRAM memory utilization and we can bring out this point in the paper revision.
>
> > Limited comparison with other NAS algorithms
>
> We compared with 8 SOTA NAS methods in our ImageNet experiment (Fig. 4a), 5 SOTA methods in the CIFAR100 experiment in Fig. 4b, and 3 SOTA methods in our super resolution experiment (Fig. 4c). Counting the additional approaches we compared UDC to in the Appendix, we compared to 15 unique SOTA algorithms in total. Therefore, we disagree that our comparison to other NAS algorithms is limited in scope.
>
> Our goal was to compare with every single NAS algorithm that: 1) produces ImageNet scale results, 2) yields models under 1.5MB, 3) yields models that can be deployed with integer math (in other words, the criteria in Table 1). We are not aware of any other NAS algorithms satisfying constraints 1-3, other than the ones we have compared with. In fact, Appendix Fig. 7-8 provides a comparison to 3 algorithms which violate 3) and we show that, even in this case, UDC is pareto-dominant.
>
> > There are other NAS papers for MCU, please also cite and compare to them. For example: https://arxiv.org/abs/2010.14246
>
> We will make sure to cite uNAS (the paper which the reviewer has referenced) as related work. At the same time, we do not see how a direct comparison between uNAS and UDC can be made. First, uNAS does not contain results for large scale datasets like ImageNet or medium-difficulty datasets like CIFAR100. Second, UDC yields results much faster: UDC finds ImageNet results in roughly 6.4 GPUD, while uNAS takes 23 GPUD to yield results on CIFAR-10. Third, UDC targets NPU deployment where compression plays a big factor, whereas uNAS targets MCUs running on traditional ARM M-class processors. Although it is not clear exactly which MCU uNAS targets, uNAS contains some experiments on NUCLEO-H743ZI2. This MCU runs on an ARM M7 processor, which does not support model compression.
>
> >What is the decompression overhead when the compressed model is executed on the target device?
>
> In the use case we studied, decompression is implemented in the Ethos U55 hardware. Since all models pass through a decompression stage during inference, there is no penalty for executing compressed models (i.e. as generated by UDC) compared to off-the-shelf models on the Ethos U55 hardware (or any hardware which has HW decompression). Moreover, for a given model, weight compression leads to memory bandwidth savings, further compounding the benefit at inference time. As we mentioned above, reducing model size allows for larger portions of weight stationary execution through compiler optimizations such as cascading (https://developer.arm.com/documentation/101888/0500/NPU-software-overview/NPU-software-tooling/The-Vela-compiler). These optimizations result in less data movement from external memory, and thus, less energy consumption and possibly lower latency.

---

> > ### Comment · Reviewer_uYMT · 2022-08-08
> > **Thanks for the response**
> >
> > Looking at the authors' response to my and other reviewers' comments. I am happy to see that my concerns are adequately addressed. I have updated my score.

---

### Official Review · Reviewer_AJPZ · 2022-07-11

**Rating:** 5
**Confidence:** 3
**Soundness:** 3 good
**Presentation:** 2 fair
**Contribution:** 2 fair

**Summary:**

The paper proposes a way of exploring compressible NNs across different architectures, sparsity levels, and quantization levels during training. The proposed framework is called Unified DNAS for Comppresssible (UDC) NNs and it can yield compressed NNs that show better accuracy-model size trade-offs than prior work.

**Questions:**

Please see the previous section.

**Limitations:**

The authors discussed the limitations of their work.

**Strengths And Weaknesses:**

The paper tackles an important problem and the proposed solution seems reasonable. The authors claim that UDC outperforms prior work in terms of accuracy-model size and the experiments seem to support this. A brief description of the prior methods that the UDC framework is compared to such as HAQ, Gong et al. [27], McDonnel, [51], FBNet, MBNetV2, Choi et al., [18] would be really helpful for readers.

In my opinion, the paper is a bit hard to follow. Specifically, Introduction could be a lot clearer with a more structured presentation and perhaps with a few subsections.

---

> ### Author Response · Authors · 2022-07-30
> **Response to Reviewer AJPZ**
>
> Thank you for the feedback. In the following, we address your concerns point by point.
>
> > The authors claim that UDC outperforms prior work in terms of accuracy-model size but I cannot find the supporting experimental results in the main paper. The authors choose to present these main experimental results in the appendix, which I find a bit strange since it is hard to evaluate the paper without the experimental results. [...] Putting the main results in the appendix is another sign of a presentation problem
>
> Our main results are shown in Fig. 4, right above line 249 in the body of the main paper. We placed the numerical results used to generate Fig. 4 in the Appendix, but we emphasize that the underlying data is exactly the same. We presented the main results in figure form for a more pleasant reading experience and recorded the numerical results to allow for easier comparison to our work.
>
> In addition, by avoiding presenting redundant information (i.e. the results in Fig. 4 in both graphical and numeric format), we were able to include additional figures that lend evidence to the merits of our method (i.e. the comparison to random search in Fig. 5a-b, the benefit of the novel number format in Fig. 5c, the ablation study in Table 3, and the illustration of deployment to NPU using the Vela compiler in Table 4).
>
> > Moreover, a brief description of the prior methods that the UDC framework is compared to such as HAQ, Gong et al. [27], McDonnel, [51], FBNet, MBNetV2, Choi et al., [18] from Table 5-6 in the appendix would be really helpful.
>
> We highlight the main differences between UDC and existing methods in Table 1, with additional explanation about how UDC differs from MCUNet, Yang et al., APQ, Gong et al.,  Choi et al., and Uhlich et al. in Section 2.
>
> We will make sure to add a more thorough explanation of existing approaches in the Appendix in the revision.

---

> > ### Comment · Reviewer_AJPZ · 2022-07-30
> > **Thanks for the response.**
> >
> > I apologize for missing Figure 4 in the main text. Figure 4 indeed justifies the authors' claims. I update my scores accordingly. I still think that the paper is a bit hard to follow, please try to make the Introduction section more clear.

---

### Meta-Review · Area_Chair_VjYU · 2022-08-22

**Recommendation:** Accept
**Confidence:** Less certain

**Metareview:**

In this paper, the authors present a new way to obtain compressible neural networks to fit on resource-constrained NPU-based hardware.

Initial reviews were mixed, but the authors successfully managed to respond to reviewers' concerns during the rebuttal period. Several reviewers pointed out clarity issues, but (1) some of these issues came from reviewers not reading the paper carefully enough, and (2) others were properly addressed by the authors. I also want to acknowledge that the most negative review is a short one, falling below NeurIPS quality standards.

After discussion, all reviewers are leaning towards acceptance, agreeing that the paper successfully demonstrates the superiority of the proposed method vs. existing relevant baselines. As a result, I also recommend acceptance.

**Award:**

No

---

### Decision · Program_Chairs · 2022-09-14

Accept